# HIV risks and recent HIV testing among transgender women in Cambodia: Findings from a national survey

**Say Sok**[1], **Reaksmey Hong**[2], **Pheak Chhoun**[1], **Navy Chann**[3], **Sovannary Tuot**[1], **Phalkun Mun**[3], **Carinne Brody**[3], **Siyan Yi** [1,4,5,6] *

1 KHANA Center for Population Health Research, Phnom Penh, Cambodia, 2 ActionAid Cambodia, Phnom Penh, Cambodia, 3 National Center for HIV/AIDS, Dermatology and STD, Phnom Penh, Cambodia, 4 Saw Swee Hock School of Public Health, National University of Singapore & National University Health System, Singapore, Singapore, 5 Center for Global Health Research, Touro University California, Vallejo, California, United States of America, 6 School of Public Health, National Institute of Public Health, Phnom Penh, Cambodia

* ephsyi@nus.edu.sg

**Data Availability Statement:** Data cannot be shared publicly because of restrictions imposed by National Center for HIV/AIDS, Dermatology and STD (NCHADS) who owns the data. Authors have

## Abstract

### Background

Globally, the prevalence of HIV among transgender women remains much higher than that of the general population, and a large proportion of them are unaware of their HIV status. Transgender women are exposed to gender-based violence and social stigma and discrimination in different settings that may create significant barriers to receiving HIV prevention and care services. This study aimed to identify factors associated with recent HIV testing among transgender women in Cambodia.

### Methods

We conducted a cross-sectional survey in 2016 among 1375 transgender women recruited from 13 provinces using a peer-based social network recruitment method. We used a structured questionnaire for face-to-face interviews and performed rapid HIV/syphilis testing onsite. We used a multiple logistic regression analysis to identify factors associated with recent HIV testing.

### Results

Of the total, 49.2% of the participants reported having an HIV test in the past six months. After controlling for other covariates, the odds of having an HIV test in the past six months was significantly lower among students (AOR 0.36, 95% CI 0.20–0.65), participants who perceived that they were unlikely to be HIV infected (AOR 0.50, 95% CI 0.32–0.78), and participants who reported always using condoms with male non-commercial partners in the past three months (AOR 0.65, 95% CI 0.49–0.85) relative to their respective reference group. The odds of having an HIV test in the past six months was significantly higher among participants who had been reached by community-based HIV services (AOR 5.01, 95% CI

no special access to the underlying data. Readers can request access to the data from NCHADS (contact via nchads@nchads.org).

**Funding:** The National Integrated Biological and Behavioral Survey among Transgender Women in Cambodia 2016 was funded by the President's Emergency Plan for AIDS Relief (PEPFAR) and the United States Agency for International Development (USAID) through the HIV/AIDS Flagship Project. The funders had no role in study design, data collection and analysis, decision to publish, or preparation of the manuscript.

**Competing interests:** The authors have declared that no competing interests exist.

3.29–7.65) and received HIV education (AOR 1.65, 95% CI 1.06–2.58) in the past six months relative to their respective reference group.

## Conclusions

Despite the widely available free HIV testing services, more than half of transgender women in this study had not received an HIV test in the past six months. Our findings suggest that a tailored and comprehensive combination prevention program, in which HIV testing is linked to care continuum and beyond, maybe an essential next step. Social media may have the potential to be promoted and utilized among transgender women populations in order to improve HIV testing and other prevention measures.

## Introduction

Transgender women have been recognized globally as one of the highest risk groups for HIV transmission and a prime target of interventions [1–4]. In 2013, the global pool prevalence of HIV among transgender women was 17.7% in low- and middle-income countries and 21.6% in high-income countries [2]. This means transgender women were about 49 times more likely to be infected with HIV than the general adult population [2]. In addition, transgender women face many health and non-health issues, including poor sexual and reproductive health, barriers in accessing health services due to stigma and discrimination, mental health problems, limited social support, and limited employment opportunities [5–8]. To address these concerns, urgent interventions have been suggested for HIV prevention and linkage to care and treatment services for transgender women globally [7,9].

Despite their needs, systematic interventions to address these issues among transgender women in most parts of the world remain inadequate. There are still shortcomings in HIV prevention and treatment programs targeting this vulnerable group. The lack of inclusion as a separate target group in the national policy response and limited application of the comprehensive prevention programs to this group are two telling examples [7,10]. HIV testing, which is the first and essential step towards the HIV care continuum, is somewhat limited, and efforts to promote the testing need to be improved [5,11]. Despite the challenges, there are instances of successful intervention programs to promote HIV testing uptake. The programs involve national policy recognition of transgender women as a key population, mobile testing, promotion of self-testing, and social and news media to promote HIV testing and other service uptake [11–14].

The global trends of the high prevalence of HIV and the myriad challenges facing transgender women have also been observed in Cambodia. The Kingdom has successfully brought down the HIV prevalence among the general population from approximately 2% in the 1990s to 0.6% in 2016 [15]. However, the prevalence remains high at 5.9% in transgender women in Cambodia in the same year [16,17], confirming the global findings that the prevalence of HIV among this population is much higher than that of the general population [2]. This rate was higher than the 4.2% prevalence in 2012 found in a similar national survey [18]. Our previous studies showed that transgender women in Cambodia also experience other issues in their daily life. They include gender-based violence, social stigma, and discrimination in different settings, substance abuse, social marginalization, and subsequent mental health problems [8,19,20]. These issues may create significant barriers to receiving regular HIV testing and early treatment. Therefore, we conducted this study to identify factors associated with recent HIV testing among transgender women in Cambodia.

## Materials and methods

### Study sites and participants

Data were collected as part of the National Integrated Biological and Behavioral Survey 2016 conducted between December 2015 and February 2016 in the capital city of Phnom Penh and 12 other provinces. These provinces were prioritized for HIV intervention programs as they had a large population of transgender women and a high HIV burden. We used a peer-based social network recruitment method to recruit the participants in six locations in Phnom Penh and 14 locations in the 12 provinces. The locations were determined based on the proportion of the required sample size and estimated population size of transgender women in each site. Individuals would be included in the study if they: (2) were 18 years or older, (2) were biologically male at birth and self-identified as female, (3) reported having sex with at least one man in the past 12 months, (5) were able to provide written consent to participate in the study, and (6) were able to communicate in Khmer, the national language of Cambodia.

During the recruitment procedures, an outreach worker from a community-based organization at each site selected four seeds (two for the age group of 18–24 years and the other two for the age group of 25 years or older). To be qualified as a seed, a participant must meet the eligibility criteria determined in the eligibility screening tool and have a well-established social network defined as knowing at least ten transgender women living within the given location. Three coupons were provided to each eligible seed to refer three transgender women for the study. Seeds would receive an incentive of US$2 for a successful referral, and expected to reach three to six recruitment waves in each site. The data collection team would recruit new seeds based on the above criteria if the enrollment halted because the initial seeds did not recruit participants or all recruitment chains had dried up.

### Training and data collection

All research team members attended a three-day data collection training, which included a review of the study protocol, confidentiality and privacy protection measures, and interview techniques. The team also rehearsed the study procedures and questionnaire administration through a pretest. Data collection team leaders conducted regular review sessions with interviewers to enable them to communicate problems that may arise during the data collection and review the progress.

We formed two data collection teams, and each team comprised of eight members. Every team included one field supervisor, one counselor, one lab technician, and five interviewers. Each field supervisor did participant eligibility screening. Each participant was assigned to a unique personal identification number used to link all data collected from each participant, but the number was not linked to any participant's identifiers. Before the data collection, each participant was requested to provide written informed consent after getting well informed about the data collection process, potential risks and benefits of participation, and their rights to decline or withdraw from the study. The participant was then interviewed in a private room with an Android tablet. Each participant received US$4 in cash for time and transport compensation and a package of three condoms.

### Questionnaire development

The questionnaire was initially developed in English and then translated into Khmer. Another translator back-translated the Khmer version to English to ensure that the content and spirit of every original item was maintained. We held consultative meetings with key stakeholders working on HIV and representatives of transgender women communities to validate the

questionnaire. The questionnaire was pretested with 20 transgender women in Phnom Penh to ensure that the wording and contents of the questionnaire were culturally acceptable and clearly understood by the participants.

The questionnaire was constructed based on items adapted from recent community-based surveys among HIV key populations in Cambodia [21,22]. Sociodemographic characteristics included study sites (urban, rural), age, marital status, main occupation, years of formal education attained, and average monthly income. Participants were also asked about gender expression and the utilization of gender-affirming hormones and surgeries. Regarding HIV risks, we collected information on sexual behaviors and condom use with different types of partners in the past three months. The questionnaire also covered experiences of symptoms of sexually transmitted infections (STIs) such as cuts or sores in the genital area, swelling in the genital area, abnormal urethral discharge, symptoms on the anus, and symptoms in the mouth or throat as well as care-seeking behaviors for the most recent symptoms.

Regarding access to community-based HIV programs, participants were asked whether they had received any services from a community-based organization in the past six months. The services included: (1) condom and lubricant distribution, (2) screening for HIV and STIs, (3) receipt of HIV and sexual reproductive health education and materials, (4) other health services such as HIV and STI treatment and referrals for other health services, (5) legal support services, and (6) online services developed for transgender women (websites, Facebook pages, hotlines).

### Data analyses

Double data entry was performed using EpiData version 3 (Odense, Denmark). We conducted bivariate analyses to compare socio-demographic characteristics, HIV risk behaviors, and access to community-based HIV programs among transgender women who had and who had not tested for HIV in the past six months. We used the Chi-square test (or Fisher's exact test when the sample sizes were smaller than five in one cell) for categorical variables and independent Student's $t$-test for continuous variables. We constructed a multiple logistic regression model to explore factors associated with recent HIV testing. We simultaneously included variables significantly associated with HIV testing in bivariate analyses at a level of $p$-value $<0.05$ in the model. We obtained adjusted odds ratios (AOR) and presented them with a confidence interval (CI) and $p$-value. We used STATA (Version 12.0 for Windows: Stata Corp, TX, USA) to conduct statistical analyses.

### Ethical considerations

This study was approved by the National Ethics Committee for Health Research (NECHR) of Ministry of Health in Cambodia (No. 420 NECHR) and FHI 360's Protection of Human Subjects Committee (PHSC No. 713897). Each participant provided written informed consent. Privacy and confidentiality were strictly protected by having the interviews conducted in a private room and removing all personal identifiers from the documents.

## Results

### Characteristics of participants

The study included 1375 transgender women with a mean age of 25.8 (SD 7.1) years. As shown in Table 1, 81.3% of the participants were from an urban community. More than two-thirds (78.1%) were never married and not living with a partner, and the mean years of formal education they had attained were nine (SD 3.4). Their main occupations included

Table 1. Comparisons of characteristics of transgender women who had and who had not tested for HIV.

| Socio-economic characteristics | | Total (*n* = 1375) | HIV testing in the past 6 months | | |
| --- | --- | --- | --- | --- | --- |
| | | | No (*n* = 699) | Yes (*n* = 676) | *p*-value* |
| Mean age (in year) | | 25.8±7.1 | 25.9±7.8 | 25.7±6.3 | 0.54 |
| Living in an urban area | | 1146 (81.3) | 590 (84.4) | 556 (82.2) | 0.28 |
| Marital Status | | | | | 0.002 |
| | Married | 7 (0.5) | 5 (0.7) | 2 (0.3) | |
| | Divorced, separated, or widowed | 18 (1.3) | 10 (1.4) | 8 (1.2) | |
| | Unmarried, not living with a partner | 1070 (78.1) | 573 (82.0) | 501 (74.1) | |
| | Unmarried, living with a partner | 260 (18.9) | 104 (14.9) | 156 (23.1) | |
| | No response | 16 (1.2) | 7 (1.0) | 9 (1.3) | |
| Mean years of formal education | | 9.0±3.4 | 9.0±3.4 | 9.0±3.4 | 0.98 |
| Main occupation | | | | | <0.001 |
| | Entertainment worker | 203 (14.8) | 102 (14.6) | 101 (14.9) | |
| | Hair dresser/beautician | 482 (35.1) | 205 (29.3) | 277 (41.0) | |
| | Office worker | 84 (6.1) | 38 (5.4) | 46 (6.8) | |
| | Laborer/farmer | 241 (17.5) | 144 (20.6) | 97 (14.3) | |
| | Self-employed | 149 (10.8) | 74 (10.6) | 75 (11.1) | |
| | Unemployed | 64 (4.7) | 35 (5.0) | 29 (4.3) | |
| | Student | 108 (7.9) | 81 (11.6) | 27 (4.0) | |
| Mean monthly income (in US$) | | 186±231.1 | 166±170 | 206±279 | 0.001 |

HIV, human immunodeficiency virus.

Values are number (%) for categorical variables and mean ± SD for continuous variables.

*Chi-square test was used for categorical variables and Student's t-test for continuous variables.

hairdressers/beauticians (35.1%), laborers/farmers (17.5%), entertainment workers (14.8%), and self-employed (10.8%). The average monthly income of the participants in the past six months was USD186 (SD 231.1).

Of the total sample, 80.5% had ever tested for HIV, and 49.2% had the test in the past six months. The facilities where they received the most recent test included a non-governmental organization (NGO) facilities (39.4%), community-based testing by outreach workers (16.7%), public facilities (15.1%), and private facilities (8.7%). The majority of those who had tested (97.5%) received the most recent test result. Only 1.2% reported currently living with HIV, and 61.2% expressed their willingness to use an HIV self-test if it were to become available. HIV testing in the past six months was significantly associated with marital status, main occupation, level of formal education, and income.

## HIV risk behaviors

Table 2 shows that 95.2% of the participants reported having had anal sexual intercourse with men, and 2.2% reported having sexual intercourse with women in the past three months. The majority (83.3%) reported a receptive role during the last anal intercourse with men. Most (81.6%) reported having had sexual intercourse with men, not in exchange for money or gifts with a mean number of non-commercial partners of 23.6 (SD 39.2) in the past three months. More than half (58.4%) reported always using condoms with non-commercial partners in the past three months. Around one-third (36.0%) reported having sexual intercourse with men in exchange for money or gifts with a mean number of commercial partners of 2.6 (SD 9.9) in the past three months. Almost two-thirds (60.0%) reported always using condoms with commercial partners in the past three months. More than two-thirds (66.2%) perceived that they were

**Table 2. Comparisons of sexual behaviors among transgender women who had and who had not tested for HIV.**

| HIV risks in the past 3 months | | Total (*n* = 1375) | HIV testing in the past 6 months | | |
|---|---|---|---|---|---|
| | | | No (*n* = 699) | Yes (*n* = 676) | *p*-value* |
| Had anal sex with men | | 1309 (95.2) | 659 (94.3) | 650 (96.2) | 0.10 |
| Main role in anal sex with men | | | | | 0.32 |
| | Insertive | 29 (2.1) | 14 (2.0) | 15 (2.2) | |
| | Receptive | 1145 (83.3) | 582 (83.3) | 563 (83.3) | |
| | Both | 135 (9.8) | 63 (9.0) | 72 (10.7) | |
| | No response | 66 (4.8) | 40 (5.7) | 26 (3.8) | |
| Had sex with male non-commercial partners | | 1122 (81.6) | 585 (83.7) | 537 (79.4) | 0.04 |
| Mean number of non-commercial male partners | | 23.6±39.2 | 21.2±39.8 | 26.0±38.5 | 0.02 |
| Always used condoms with male non-commercial partners | | 703 (58.4) | 381 (65.0) | 322 (52.2) | <0.001 |
| Had sex with male commercial partners | | 495 (36.0) | 242 (34.6) | 253 (37.4) | 0.19 |
| Mean number of male commercial partners | | 2.6±9.9 | 2.3±9.3 | 2.9±10.4 | 0.33 |
| Always used condoms with male commercial partners | | 246 (60.0) | 113 (57.1) | 133 (62.7) | 0.24 |
| Had an STI symptom | | 193 (14.0) | 95 (13.6) | 98 (14.5) | 0.63 |
| Perceived likelihood of HIV infection | | | | | 0.02 |
| | Very likely | 153 (11.1) | 62 (8.9) | 91 (13.5) | |
| | Likely | 758 (55.1) | 387 (55.4) | 371 (54.9) | |
| | Unlikely | 345 (25.1) | 191 (27.3) | 154 (22.8) | |
| | Very unlikely | 119 (8.7) | 59 (8.4) | 60 (8.9) | |

HIV, human immunodeficiency virus; STI, sexually transmitted infections.

Values are number (%) for categorical variables and mean ± SD for continuous variables.

*Chi-square test was used for categorical variables and Student's t-test for continuous variables.

"likely" or "very likely" to be infected with HIV. HIV testing in the past six months was significantly associated with a higher number of non-commercial male partners, having sexual intercourse with non-commercial male partners, and always using condoms with non-commercial partners in the past three months. HIV testing in the past six months was also significantly associated with the perceived likelihood of getting HIV infection.

## Access to community-based HIV services

Of the total, 45.0% of the participants had been reached by at least one form of community-based HIV services in the past six months. The main services included HIV and STI education (35.6%), condom distribution (38.0%), lubricant distribution (31.2%), and general online HIV services (35.1%). Participants received HIV-related information through websites (7.0%), Facebook pages (10.8%), voice messages from Voice4U (an interactive voice response system that provides education and counseling services and on-demand HIV and other health-related information) (5.5%), and Voice4U calls (5.7%). HIV testing in the past six months was significantly associated with having access to all forms of community-based services in the past six months (Table 3).

## Factors associated with recent HIV testing

Table 4 shows factors associated with recent HIV testing in a multiple logistic regression model. After controlling for other covariates, the odds of having an HIV test in the past six

**Table 3. Comparisons of access to community-based HIV services among transgender women who had and who had not tested for HIV.**

| Community-based HIV services in the past 6 months | Total (*n* = 1375) | HIV testing in the past 6 months | | |
| --- | --- | --- | --- | --- |
| | | No (*n* = 699) | Yes (*n* = 676) | *p*-value[*] |
| Reached by community-based HIV services | 619 (45.0) | 155 (22.2) | 464 (68.6) | <0.001 |
| HIV-related information and education | 490 (35.6) | 115 (16.5) | 375 (55.5) | <0.001 |
| Condom distribution | 522 (38.0) | 136 (19.5) | 386 (57.1) | <0.001 |
| Lubricant distribution | 429 (31.2) | 124 (17.7) | 305 (45.1) | <0.001 |
| Facebook pages MSM/transgender women | 149 (10.8) | 38 (5.4) | 111 (16.4) | <0.001 |
| Websites transgender women | 96 (7.0) | 23 (3.3) | 73 (10.8) | <0.001 |
| Voice messages or calling Voice4U[†] | 76 (5.5) | 19 (2.7) | 57 (8.4) | <0.001 |
| Legal aid services | 47 (3.4) | 12 (1.7) | 35 (5.7) | <0.001 |

HIV, human immunodeficiency virus; MSM, men who have sex with men.

Values are number (%) of respondents.

[*]Chi-square test was used.

[†]Voice4U is an interactive voice response system that provides on-line counseling, on-demand HIV, and health-related information to HIV key populations.

**Table 4. Factors associated with recent HIV testing among transgender women in a multiple logistic regression model (*n* = 1375).**

| Variables in the model | | HIV testing in the past 6 months | |
| --- | --- | --- | --- |
| | | Unadjusted OR (95% CI) | Adjusted OR (95% CI) |
| Main occupation | | | |
| | Entertainment worker | Reference | Reference |
| | Hair dresser/beautician | 1.36 (0.98–1.89) | 1.34 (0.92–1.95) |
| | Office worker | 0.60 (0.37–0.98)[*] | 0.93 (0.51–1.67) |
| | Laborer/farmer | 0.68 (0.47–1.99) | 0.81 (0.43–1.52) |
| | Self-employed | 1.02 (0.67–1.56) | 1.02 (0.63–1.66) |
| | Unemployed | 0.84 (0.48–1.47) | 0.96 (0.51–1.83) |
| | Student | 0.33 (0.20–0.56)[***] | 0.36 (0.20–0.65)[**] |
| Perceived likelihood of HIV infection | | | |
| | Very likely | Reference | Reference |
| | Likely | 0.65 (0.46–0.93)[*] | 0.86 (0.37–1.84) |
| | Unlikely | 0.55 (0.37–0.81)[**] | 0.50 (0.32–0.78)[**] |
| | Very unlikely | 0.69 (0.43–1.12) | 0.59 (0.04–1.03) |
| Condom use with male non-commercial in past 3 months | | | |
| | Not always | Reference | Reference |
| | Always | 0.58 (0.40–0.82)[**] | 0.65 (0.49–0.85)[**] |
| Reached by community-based HIV services in the past 6 months | | | |
| | No | Reference | Reference |
| | Yes | 2.04 (1.41–2.95)[***] | 2.01 (1.29–3.65)[***] |
| Received HIV education and materials in the past 6 months | | | |
| | No | Reference | Reference |
| | Yes | 2.22 (1.52–3.26)[***] | 1.65 (1.06–2.58)[*] |

Abbreviations: CI, confidence interval; HIV, human immunodeficiency virus; OR, odds ratio.

Variables in the table were the ones that remained statistically significant after several steps of model fitting.

[*]$p < 0.05$

[**]$p < 0.01$

[***]$p < 0.001$.

months was significantly lower among students (AOR 0.36, 95% CI 0.20–0.65), participants who perceived that they were unlikely to be HIV infected (AOR 0.50, 95% CI 0.32–0.78), and participants who reported always using condoms with male non-commercial partners in the past three months (AOR 0.65, 95% CI 0.49–0.85) relative to their respective reference group. The odds of having an HIV test in the past six months was significantly higher among participants who had been reached by community-based HIV services (AOR 5.01, 95% CI 3.29–7.65) and received HIV education (AOR 1.65, 95% CI 1.06–2.58) in the past six months relative to their respective reference group.

## Discussion

Despite the widely available free HIV testing services across Cambodia, one in five transgender women in this national survey had never tested for HIV in their lifetime. About half had not been tested in the past six months. This low rate is somewhat concerning and calls for more rigorous efforts to promote HIV testing among this key population. The results indicate that transgender women who are students, who perceived that they were unlikely to have HIV, and who reported always using condoms with male non-commercial partners were less likely to have the test. Transgender women who had been reached by community-based HIV services and who had received HIV education were more likely to have a recent HIV test. This may be attributed to the positive contribution of the grassroots approaches to HIV interventions in Cambodia.

To the best of our knowledge, there have been no other studies that examine factors associated with HIV testing among transgender women in Cambodia. Studies from other countries similarly reported consistently low rates of HIV testing among transgender women. A study in Thailand, for example, showed that 53.4% of transgender women had never tested for HIV [23]. In the United States, Schulden et al. reported that 8.1% of transgender women recruited from Florida, New York City, and San Francisco had never received HIV testing in their lifetime, and 46.5% did not have it in the past year [24]. These persistently low rates may be explained by their limited access to HIV testing services due to factors in different domains, including socioeconomic marginalization pounded by political and legal pressures and the limited trans-specific services [2,25]. Given the high prevalence of HIV found in this national survey [16,17] and two other previous studies in Cambodia [26,27], the low HIV testing rate can be a public health concern and needs to be appropriately addressed.

In our multivariable regression analysis, we found that transgender women who were students were less likely to have had a recent HIV test. The low HIV testing rate among students could have happened because they were yet to be sexually active and perceived that they were at low risk of HIV infection. Our recent study on 1359 undergraduate students in two universities in Cambodia, for example, showed that only one in 10 students engaged in sexual intercourse in the past 12 months, of whom 42% did not use a condom in the last intercourse [28]. Moreover, as students were not the main target of HIV intervention programs in Cambodia, their awareness of the importance for the testing could be limited. Studies elsewhere on students in general and transgender students in particular also reported low rates of HIV testing, and they attributed this phenomenon to fear, anxiety, stigma, and discrimination in receiving the test [29–32]. Since young students may later adopt risky sexual behaviors and are thus at risk of HIV infection [33,34], promoting HIV test uptake among them is necessary. In this regard, efforts are required to mainstream lessons on HIV and STIs into the curriculum and extra-curricular activities as a preventive measure and to mobilize for HIV testing in particular.

In this study, transgender women who perceived that they were unlikely to have HIV infection and who always used condoms with male non-commercial partners were less likely to

have the test. This study did not capture how and why these attributes are related to not having an HIV test; hence, further research is needed to better understand the correlations. This may happen simply because these individuals had safer sexual practices and were more confident that they were unlikely to get HIV and other STIs. On the other hand, other studies found that the fear of HIV testing outcomes, related stigma [30,32], and the general tendency among transgender communities in believing that they were not at risk for HIV were accountable for not taking the test [31,35].

Another key finding from our study is that not-for-profit facilities were the most common location for obtaining HIV testing and counseling services. This preference is confirmed by previous studies in Cambodia and elsewhere, which indicate that community-based services are preferred by transgender women and other key populations such as female entertainment workers and men who have sex with men [20–22,24]. The effectiveness of community-based programs for HIV prevention and treatment, including testing, is well documented in global literature. The positive outcomes of the intervention programs in Indonesia and India in improving the quality of life of the recipients are two positive illustrations [6,36]. Some factors that may contribute to transgender women's preference for not-for-profit facilities include the presence of trans-specific and friendly organizations, targeted outreach activities, and recommendations from peers [6,7,36]. Thus, any endeavor to promote HIV prevention, including HIV testing, may take advantage of the community-based approaches.

In this study, transgender women who had a recent HIV test were more likely to use Facebook pages for men who have sex with men and transgender populations. However, the overall level of Facebook usage was low. Global literature confirms the effectiveness of social media in increasing HIV testing uptake and safe sexual behaviors among HIV key populations, including transgender women. Challenges remain to be addressed, including a low level of social media literacy and proper design of the media tools and strategies. Such media can enhance social support and reduce social exclusion, psychosocial pressure, and stigma [14,37–39]. Given the rising popularity of social media as a tool for communication and dissemination [20,41], the low consumption of the customized social media platforms such as voice messages and interactive voice response systems need to be better understood to promote the usage among transgender communities [40]. Efforts are needed to investigate the impact of social media on HIV prevention and access to HIV services. Findings from such investigations would support the design of effective media tools and strategies to curb HIV prevention and linkage to treatment and care services.

This study has several limitations. First, we conducted the survey in only the capital city and 12 provinces with a high burden of HIV and large transgender women population size. This selection may limit the generalizability of the findings to other provinces where HIV knowledge, attitude, and practices may differ. Second, the study was cross-sectional by design, and hence it was not possible to capture the trends and changes over time and causal relationships. Third, initial seeds were recruited by community-based NGOs, who may have recruited participants who received HIV interventions from them. Fourth, the survey depended on participants' self-reporting, which may create social desirability bias, especially on sensitive issues. Despite these shortcomings, this study contributes to a better understanding of the factors associated with HIV testing among transgender women in Cambodia, which is quite limited in the scientific literature.

## Conclusions

This study is one of the few studies that examine factors associated with HIV testing among transgender women in low- and middle-income countries. The findings suggest that

transgender women who were students, who perceived that they were unlikely to have HIV, and who do not use condoms consistently should be the main targets of HIV testing and prevention efforts. Tailored and comprehensive combination prevention programs, in which HIV testing is linked to the care continuum and beyond, may be necessary. Social media can be promoted and utilized among transgender women populations to improve HIV testing uptake and other prevention measures. In this regard, the low use of social media such as Facebook needs to be better understood. Utilizing effective strategies to ensure that this type of outreach and information dissemination can help in the fight against HIV. More studies are needed to develop innovative interventions to reach approximately 20% of transgender sub-populations who have never been reached by traditional HIV testing strategies. The promotion of HIV testing should be conducted within a comprehensive HIV program in which HIV testing is combined with other transgender-friendly prevention, care, and treatment services. The program may include HIV self-testing and pre-exposure prophylaxis (PrEP), which have been found to be well accepted by HIV key populations, including transgender women in Cambodia [13,41].

## Acknowledgments

The National Integrated Biological and Behavioral Survey among Transgender Women in Cambodia 2016 was technically led by the National Center for HIV/AIDS, Dermatology and STD. We thank all data collection team members, community-based organizations, community support volunteers, and all participants for their contribution to this survey.

## Author Contributions

**Conceptualization:** Say Sok, Reaksmey Hong, Pheak Chhoun, Navy Chann, Sovannary Tuot, Phalkun Mun, Carinne Brody, Siyan Yi.

**Data curation:** Say Sok, Reaksmey Hong, Carinne Brody, Siyan Yi.

**Formal analysis:** Say Sok, Reaksmey Hong, Carinne Brody, Siyan Yi.

**Funding acquisition:** Sovannary Tuot, Phalkun Mun, Siyan Yi.

**Investigation:** Say Sok, Reaksmey Hong, Pheak Chhoun, Navy Chann, Sovannary Tuot, Phalkun Mun, Carinne Brody, Siyan Yi.

**Methodology:** Say Sok, Reaksmey Hong, Pheak Chhoun, Navy Chann, Sovannary Tuot, Phalkun Mun, Carinne Brody, Siyan Yi.

**Project administration:** Pheak Chhoun, Navy Chann, Sovannary Tuot, Phalkun Mun, Siyan Yi.

**Resources:** Navy Chann, Sovannary Tuot, Phalkun Mun, Siyan Yi.

**Software:** Say Sok, Reaksmey Hong, Carinne Brody, Siyan Yi.

**Supervision:** Pheak Chhoun, Navy Chann, Sovannary Tuot, Phalkun Mun, Siyan Yi.

**Validation:** Say Sok, Reaksmey Hong, Carinne Brody, Siyan Yi.

**Visualization:** Say Sok, Reaksmey Hong, Carinne Brody, Siyan Yi.

**Writing – original draft:** Say Sok, Reaksmey Hong, Carinne Brody, Siyan Yi.

**Writing – review & editing:** Say Sok, Reaksmey Hong, Pheak Chhoun, Navy Chann, Sovannary Tuot, Phalkun Mun, Carinne Brody, Siyan Yi.

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
