## [Decision Letter · Decision Letter 0]

5 May 2020

PONE-D-20-08627

HIV risks and recent HIV testing among transgender women in Cambodia: Findings from a national survey using respondent driven method

PLOS ONE

Dear Dr. Yi,

Thank you for submitting your manuscript to PLOS ONE. After careful consideration, we feel that it has merit but does not fully meet PLOS ONE’s publication criteria as it currently stands. Therefore, we invite you to submit a revised version of the manuscript that addresses the points raised during the review process.

Both reviewers are supportive of the manuscript but note that the description of the study as using RDS methodology is not accurate and should be revised accordingly (perhaps as "RDS-informed" methods). Other comments regarding characterization of sexual behavior and HIV risk are below.

We would appreciate receiving your revised manuscript by Jun 19 2020 11:59PM. To enhance the reproducibility of your results, we recommend that if applicable you deposit your laboratory protocols in protocols.io, where a protocol can be assigned its own identifier (DOI) such that it can be cited independently in the future. For instructions see: http://journals.plos.org/plosone/s/submission-guidelines#loc-laboratory-protocols

We look forward to receiving your revised manuscript.

Kind regards,

Jesse L Clark, MD, MSc

Academic Editor

PLOS ONE

Journal Requirements:

'The National Integrated Biological and Behavioral Survey among Transgender Women in Cambodia 2016 was technically led by the National Center for HIV/AIDS, Dermatology and STD and funded by the President's Emergency Plan for AIDS Relief (PEPFAR) and the United States Agency for International Development (USAID) through the HIV/AIDS Flagship Project.'

'The funders had no role in study design, data collection and analysis, decision to publish, or preparation of the manuscript.'

Reviewers' comments:

Reviewer's Responses to Questions

**Comments to the Author**

1. Is the manuscript technically sound, and do the data support the conclusions?

Reviewer #1: Partly

Reviewer #2: Yes

2. Has the statistical analysis been performed appropriately and rigorously? 

Reviewer #1: Yes

Reviewer #2: Yes

3. Have the authors made all data underlying the findings in their manuscript fully available?

Reviewer #1: No

Reviewer #2: No

4. Is the manuscript presented in an intelligible fashion and written in standard English?

Reviewer #1: Yes

Reviewer #2: Yes

5. Review Comments to the Author

Reviewer #1: While this paper presents important information about a population at high risk for HIV infection and is well written I have a concern about over representing methodological rigour by saying the study used RDS. While the study did select seeds and used recruitment coupons in 13 areas this is not an RDS study. RDS studies cannot be conducted as a whole across geographies where there are no recruitment connections between geographies. There is also no data presented on recruitment even within individual sites. So it might be more accurate to say, and I strongly encourage this, this study used peer based social network recruitment which was initiated by outreach workers.

Reviewer #2: This paper presents relevant information about transgender women in Cambodia that can inform prevention interventions for a vulnerable community.

General:

The study has not used RDS as part of the design: RDS implies sampling driven by respondents (all belonging to a single network across sites) and weighed analysis of the data adjusting for social network size and characteristics of recruitment. In this case, without the weighed analysis, the design is a cross sectional study using snowball or chain referral as sampling methodology. RDS implies sampling and analysis.

Abstract:

On the background section authors state: “Cambodian transgender women are exposed to several issues in their daily life that may create significant barriers to receiving HIV prevention”. They should give examples of those issues to avoid vague phrasing.

Methods: see above regarding RDS.

Results

Authors present results about the main outcome at first, then describe the population included in the study while at the same time stratifying according to the main outcome (as shown in table 2 and on last paragraph of page 7). This way of presenting results might be a bit confusing for the reader. I suggest that authors first present the socio-demographic characteristics of the population included in the study and then carry out with the stratification of results regarding the main outcome. In that same line they could reconsider the order of tables.

Regarding the interpretation of the model, authors should review their phrasing when analyzing ORs associated with the outcome variable (page 11). For example: “participants who had been tested for HIV in the past six months remained significantly less likely to be a student (AOR= 0.36, 95% CI= 0.20-0.65)” (page 11). Though the description is taken literally from the table it is hard to contextualize. Authors should consider rephrasing, stating the condition (being a student or not) associated with the outcome: (less probability of having an HIV test in the past six months or vice versa).

Table 5: Authors could include unadjusted and adjusted ORs.

Discussion:

Though the outcome of the study is HIV testing in the past six months, there is a considerable group that has never been tested. Analyzing that information may be a matter of a different paper. However, that group should be acknowledged when formulating recommendations for testing interventions.

Also, when stating prevention interventions, testing should not be seen as an isolated intervention. Enhanced access to testing should be combined with other available interventions (eg PrEP, access to condoms/lubricants, early initiation of treatment).

The recommendations stated could be stronger if authors discuss testing with other available options for transgender women in Cambodia.

Page 13: Authors state, “The low HIV testing rate among students could have happened because they were yet to be sexually active or that they were not a main target of intervention programs, and hence their awareness of the importance for the testing is limited”. Authors have the data to check for the first assumption (sexual activity).

6. PLOS authors have the option to publish the peer review history of their article (what does this mean?). If published, this will include your full peer review and any attached files.

Reviewer #1: No

Reviewer #2: No

---

## [Author Response · Author response to Decision Letter 0]

16 Jun 2020

1. Please ensure that your manuscript meets PLOS ONE's style requirements, including those for file naming. The PLOS ONE style templates can be found at:

and

RESPONSE: Thank you for the reminder. We have ensured that the manuscript meets PLOS ONE’s style and requirements, including file naming.

RESPONSE: Data used for this study were conducted as part of the National Integrated Biological and Behavioral Survey among Transgender Women 2016. The survey was led and data owned by the National Center for HIV/AIDS, Dermatology and STD (NCHADS). As an NGO partner, our roles were to provide technical support, and we do not have rights to make the data publicly available without the permission from NCHADS.

'The National Integrated Biological and Behavioral Survey among Transgender Women in Cambodia 2016 was technically led by the National Center for HIV/AIDS, Dermatology and STD and funded by the President's Emergency Plan for AIDS Relief (PEPFAR) and the United States Agency for International Development (USAID) through the HIV/AIDS Flagship Project.'

'The funders had no role in study design, data collection and analysis, decision to publish, or preparation of the manuscript.'

a. Please clarify the sources of funding (financial or material support) for your study. List the grants or organizations that supported your study, including funding received from your institution.

d. If you did not receive any funding for this study, please state: “The authors received no specific funding for this work.”

RESPONSE: We have removed funding-related text from the manuscript. We would like to update our Funding Statement as follows:

“The National Integrated Biological and Behavioral Survey among Transgender Women in Cambodia 2016 was funded by the President's Emergency Plan for AIDS Relief (PEPFAR) and the United States Agency for International Development (USAID) through the HIV/AIDS Flagship Project. The funders had no role in study design, data collection and analysis, decision to publish, or preparation of the manuscript. No specific funding was available for this analysis and publishing the manuscript.”

Reviewer 1

While this paper presents important information about a population at high risk for HIV infection and is well written I have a concern about over representing methodological rigour by saying the study used RDS. While the study did select seeds and used recruitment coupons in 13 areas this is not an RDS study. RDS studies cannot be conducted as a whole across geographies where there are no recruitment connections between geographies. There is also no data presented on recruitment even within individual sites. So it might be more accurate to say, and I strongly encourage this, this study used peer based social network recruitment which was initiated by outreach workers.

RESPONSE: We thank reviewer for your kind contribution to this manuscript. We agree with this important point and have revised the wording throughout the paper. Where suitable, ‘peer-based social network recruitment’ method has been used.

Reviewer 2

This paper presents relevant information about transgender women in Cambodia that can inform prevention interventions for a vulnerable community.

General:

The study has not used RDS as part of the design: RDS implies sampling driven by respondents (all belonging to a single network across sites) and weighed analysis of the data adjusting for social network size and characteristics of recruitment. In this case, without the weighed analysis, the design is a cross sectional study using snowball or chain referral as sampling methodology. RDS implies sampling and analysis.

RESPONSE: We thank reviewer for your kind contribution to this manuscript and apologize for the oversight. This comment was also raised by another reviewer. We agree with this important point and have revised the wording throughout the paper, including removing it from the title. Where suitable, ‘peer-based social network recruitment’ method has been used.

Abstract:

On the background section authors state: “Cambodian transgender women are exposed to several issues in their daily life that may create significant barriers to receiving HIV prevention”. They should give examples of those issues to avoid vague phrasing.

RESPONSE: We have revised the Background of the Abstract as follows: “Transgender women are exposed to gender-based violence as well as social stigma and discrimination in different settings that may create significant barriers to receiving HIV prevention and care services.” Line 22-24.

Results

Authors present results about the main outcome at first, then describe the population included in the study while at the same time stratifying according to the main outcome (as shown in table 2 and on last paragraph of page 7). This way of presenting results might be a bit confusing for the reader. I suggest that authors first present the socio-demographic characteristics of the population included in the study and then carry out with the stratification of results regarding the main outcome. In that same line they could reconsider the order of tables.

Regarding the interpretation of the model, authors should review their phrasing when analyzing ORs associated with the outcome variable (page 11). For example: “participants who had been tested for HIV in the past six months remained significantly less likely to be a student (AOR= 0.36, 95% CI= 0.20-0.65)” (page 11). Though the description is taken literally from the table it is hard to contextualize. Authors should consider rephrasing, stating the condition (being a student or not) associated with the outcome: (less probability of having an HIV test in the past six months or vice versa).

RESPONSE: We have moved the results around to start with socio-demographic characteristics of the participants followed by the characteristics of HIV testing and comparisons of the socio-demographic characteristics among participants who had and who had not been tested for HIV in the past six months. We have also revised the interpretation of the multiple regression model results as follows (143-250): “After controlling for other co-variates, the odds of having an HIV test in the past six months was significantly lower among students (AOR 0.36, 95% CI 0.20-0.65), participants who perceived that they were unlikely to be HIV infected (AOR 0.50, 95% CI 0.32-0.78), and participants who reported always using condoms with male non-commercial partners in the past three months (AOR 0.65, 95% CI 0.49-0.85) relative to their respective reference group. The odds of having an HIV test in the past six months was significantly higher among participants who had been reached by community-based HIV services (AOR 5.01, 95% CI 3.29-7.65) and received HIV education (AOR 1.65, 95% CI 1.06-2.58) in the past six months relative to their respective reference group.” We have also revised results in the Abstract accordingly. Lines 36-44.

Table 5: Authors could include unadjusted and adjusted ORs.

RESPONSE: We have added unadjusted odds ratio to the table as suggested. To simplify the table, we have removed p-value column and indicated the significance levels in the footnotes of the table.

Discussion:

Though the outcome of the study is HIV testing in the past six months, there is a considerable group that has never been tested. Analyzing that information may be a matter of a different paper. However, that group should be acknowledged when formulating recommendations for testing interventions.

RESPONSE: We are grateful to the reviewer for this very useful comment. We have now extended the recommendations to emphasize the need of more research to explore innovative strategies to reach transgender sub-populations who have never been reached by the current programs for HIV testing: “More studies are needed to develop innovative interventions to reach the approximately 20% of transgender sub-populations who have never been reached by the traditional strategies for HIV testing. The promotion of HIV testing should be conducted within a comprehensive HIV program, in which HIV testing is combined with other transgender-friendly prevention, care, and treatment services.” Lines 350-354.

Also, when stating prevention interventions, testing should not be seen as an isolated intervention. Enhanced access to testing should be combined with other available interventions (eg PrEP, access to condoms/lubricants, early initiation of treatment). The recommendations stated could be stronger if authors discuss testing with other available options for transgender women in Cambodia.

RESPONSE: We are thankful to the reviewer for this very important comment. We have addressed this point by extending the recommendations to also include a suggestion for combining promotion of HIV testing in a comprehensive intervention program that also includes other prevention and care services such as PrEP and HIV self-testing that have been found to be well accepted in key populations in low- and middle-income countries including transgender women in Cambodia based on findings from our previous studies. Lines 355-357.

Page 13: Authors state, “The low HIV testing rate among students could have happened because they were yet to be sexually active or that they were not a main target of intervention programs, and hence their awareness of the importance for the testing is limited”. Authors have the data to check for the first assumption (sexual activity).

RESPONSE: We have now added findings from our previous study to support this statement: “The low HIV testing rate among students could have happened because they were yet to be sexually active and perceived that they were at low risk of HIV infection. Our recent study on 1359 undergraduate students in two universities in Cambodia, for example, showed that only one in 10 students engaged in sexual intercourse in the past 12 months; of whom 42% did not use a condom in the last intercourse.” Lines 282-286.

---

## [Editor Report · Decision Letter 1]

10 Aug 2020

PONE-D-20-08627R1

HIV risks and recent HIV testing among transgender women in Cambodia: findings from a national survey

PLOS ONE

Dear Dr. Yi,

Thank you for submitting your manuscript to PLOS ONE. After careful consideration, we feel that it has merit but does not fully meet PLOS ONE’s publication criteria as it currently stands. Therefore, we invite you to submit a revised version of the manuscript that addresses the points raised during the review process.

I

I apologize for the extended delay in reviewing your revised manuscript. Due to the COVID-19 pandemic, reviewer availability has been extremely limited and neither of the original reviewers were available to read the revised manuscript. Based on my review, the manuscript is ready for publication after one minor correction: On Page 8, please change the subheading from "HIV risky behaviors" to "HIV risk behaviors."

We look forward to receiving your revised manuscript.

Kind regards,

Jesse L Clark, MD, MSc

Academic Editor

PLOS ONE

---

## [Author Response · Author response to Decision Letter 1]

11 Aug 2020

We thank the editor for your kind consideration on our manuscript. In response to your minor comment, we have now revised the manuscript accordingly. We also took this opportunity to proofread the manuscript. We have removed typos and errors, changed passive to active voice where suitable, and rewrote long sentences for improving readability.

---

## [Editor Report · Decision Letter 2]

14 Aug 2020

HIV risks and recent HIV testing among transgender women in Cambodia: findings from a national survey

PONE-D-20-08627R2

Dear Dr. Yi,

We’re pleased to inform you that your manuscript has been judged scientifically suitable for publication and will be formally accepted for publication once it meets all outstanding technical requirements.

Kind regards,

Jesse L Clark, MD, MSc

Academic Editor

PLOS ONE
---

## [Editor Report · Acceptance letter]

18 Aug 2020

PONE-D-20-08627R2 

HIV risks and recent HIV testing among transgender women in Cambodia: findings from a national survey 

Dear Dr. Yi:

I'm pleased to inform you that your manuscript has been deemed suitable for publication in PLOS ONE. Congratulations! Your manuscript is now with our production department. 

Kind regards, 

on behalf of

Dr. Jesse L Clark 

Academic Editor

PLOS ONE